# Early-Transmitted Variants and Their Evolution in a HIV-1 Positive Couple: NGS and Phylogenetic Analyses

**DOI:** 10.3390/v13030513

**Published:** 2021-03-19

**Authors:** Alessia Lai, Vania Giacomet, Annalisa Bergna, Gian Vincenzo Zuccotti, Gianguglielmo Zehender, Mario Clerici, Daria Trabattoni, Claudio Fenizia

**Affiliations:** 1Department of Biomedical and Clinical Sciences, University of Milan, Via G.B. Grassi 74, 20157 Milan, Italy; alessia.lai@unimi.it (A.L.); bergna.anna@gmail.com (A.B.); gianguglielmo.zehender@unimi.it (G.Z.); daria.trabattoni@unimi.it (D.T.); 2Clinic of Pediatrics, ASST Fatebenefratelli-Sacco, Sacco Clinical Sciences Institute, Via G.B. Grassi 74, 20157 Milan, Italy; vania.giacomet@unimi.it (V.G.); GianVincenzo.Zuccotti@unimi.it (G.V.Z.); 3Department of Pathophysiology and Transplantation, University of Milan, Via F. Sforza 35, 20122 Milan, Italy; mario.clerici@unimi.it; 4IRCCS Fondazione Don Carlo Gnocchi, Via Capecelatro 66, 20148 Milan, Italy

**Keywords:** human immunodeficiency virus (HIV), HIV T/F variants, HIV deep phylogenetic NGS analyses, HIV quasispecies, mucosal bottleneck, HIV evolution, HIV sexual transmission

## Abstract

We had access to both components of a couple who became infected with human immunodeficiency virus (HIV)-1 through sexual behavior during the early initial phase of infection and before initiation of therapy. We analyzed blood samples obtained at the time of diagnosis and after six months of combined antiretroviral therapy. Next-generation sequencing (NGS) and phylogenetic analyses were used to investigate the transmission and evolution of HIV-1 quasispecies. Phylogenetic analyses were conducted using Bayesian inference methods. Both partners were infected with an HIV-1 B subtype. No evidence of viral recombination was observed. The lowest intrapersonal genetic distances were observed at baseline, before initiation of therapy, and in particular in the V1V2 fragment (distances ranging from 0.102 to 0.148). One HIV-1 single variant was concluded to be dominant in all of the HIV-1 regions analyzed, although some minor variants could be observed. The same tree structure was observed both at baseline and after six months of therapy. These are the first extended phylogenetic analyses performed on both members of a therapy-naïve couple within a few weeks of infection, and in which the effect of antiretroviral therapy on viral evolution was analyzed. Understanding which HIV-1 variants are most likely to be transmitted would allow a better understanding of viral evolution, possibly playing a role in vaccine design and prevention strategies.

## 1. Introduction

During the course of the infection, human immunodeficiency virus (HIV)-1 typically accumulates a high level of genetic diversity [1]. This high mutation rate allows the virus to escape immune recognition and to develop resistance to antivirals, while adapting to the host [2]. The HIV genome spontaneously accumulates mutations, as the reverse transcriptase lacks the proofreading activity, and RNA editing enzymes, such as those in the APOBEC3 family, profoundly operate on the HIV-1 genome [3,4,5,6]. Therefore, each round of HIV-1 infection generates new mutations, which accumulate in new variants and quasispecies, continuously refining the viral fitness for that particular host, tissue or cell tropism [2]. Indeed, the generation of such genetic diversity is a strong advantage for HIV-1 to infect new hosts.

During the last decade it has been established that in the vast majority of primary infections the host is colonized by only one out of potentially thousands of HIV-1 variants [7]. In fact, in 2008, Keele and colleagues first described that 80% of cases of primary HIV-1 infection are founded by one single variant, named transmitted/founder (T/F) variant, while in the remaining 20% of cases two to five distinct T/F variants can be observed [8]. These observations were later confirmed [9,10,11,12].

T/F variants are modeled by both stochastic and selective pressures, and the mucosal bottleneck plays a major role in this process [7]. Thus, in individuals chronically infected with HIV-1, an extremely high genetic diversity accumulates in the blood compartment, but just a few clonally and randomly selected expanded variants are observed in the genital tract [7,13]. After the penetration into the recipient’s genital tract, one or few variants alone are actually able to reach the recipient’s blood compartment and these are usually the best-fitting and fastest-replicating variants [14,15]. These T/F variants are initially characterized by a very low heterogeneity and they drive the acute infection in the new host [7,16]. Gradually, HIV-1 genetic diversity is then shaped during chronic infection. Interestingly, those individuals who are infected by multiple variants have a higher chance of displaying a higher viral load set-point one year after HIV-1 diagnosis [17]. Based on experimental evidence, it has been suggested that the transmission of multiple T/F variants is not associated with features of the variants themselves, but rather with yet unspecified traits of the host [17,18].

Successful combined antiretroviral therapy (cART) leads to the suppression of viral replication, and HIV becomes undetectable in peripheral blood. However, HIV-1 hides in resting memory CD4+ T cells as a latent reservoir [19,20]. As soon as the conditions are more favorable for HIV-1 (i.e., therapy interruption), viral replication rebounds; in this case infection is driven by rebounding/founder (R/F) variants in lymphoid tissues [21] and multiple R/F variants are typically involved. Moreover, integrated HIV-1 proviruses can be replicated by clonally expanded CD4+ T cells [22,23], similar to what is observed in other chronic retroviral infections such as HTLV-1 [24,25,26].

In roughly 25% of HIV-1-positive patients undergoing cART, the virus reservoir increases over time despite treatments. Viral blips, which result in transient detectable viremia, are associated with a reservoir increase and/or a slower reservoir decay [27]. Recent data showed that, in treated HIV-1-infected subjects, intact and defective provirus declined over time, with a more rapid decline in the first years of cART and a faster decay in cells with intact genomes [28,29,30]. Latently HIV-1-infected cells can occasionally be activated by specific immunological antigen recognition, or as a consequence of a bystander effect during local inflammation; this results in the triggering of HIV-1 replication [31]. During viral blips, multiple and heterogeneous variants can be detected. Low-level viremia and/or replication burst during the blips might contribute to HIV-1 genetic diversity and shape the pool of viral variants and the reservoir. A highly expanded clone of HIV-infected CD4+ T cells, which produced the infectious virus at a level that caused persistent plasma viremia, was identified. This result shows that cells containing replication-competent HIV-1 proviruses can clonally expand and persist in vivo [23]. On the other hand, it is now well established that HIV-1 is extremely unlikely to be transmitted when it is undetectable in the blood (U = U, undetectable = untransmittable) [32,33,34,35,36,37,38].

A better understanding of which variants are most likely to be found in a new infection could help vaccine design and prevention strategies. Moreover, since the initial phase of HIV-1 infection directly impacts the progression of the disease [17], future approaches that limit the number and the heterogeneity of T/F variants could become promising therapeutic avenues.

We characterized T/F variants in a young couple who became infected through heterosexual activity. As the phylogenetic analyses represent the gold standard in the study of HIV evolution and transmission, we adopted the Bayesian inference methods to study the variants transmitted between the two individuals in four viral genomic regions. This is the first study performing a phylogenetic analysis of the T/F variants in a therapy-naïve couple within a few weeks of infection and after the initiation of antiretroviral therapy.

## 2. Materials and Methods

### 2.1. Patients’ Description

A fourteen-year-old female (M) was admitted on 16th June 2017 to the pediatric department at Luigi Sacco Hospital (Milan, Italy) for a 6-day persistent fever associated with pharyngodynia. During the hospitalization, blood was drawn in order to perform routine tests. As the HIV test result was positive, an immuno-virological evaluation was initiated. M reported having a sexual relationship with a fifteen-year-old male (G). G was visited at our pediatric infectious disease clinic and underwent immuno-virological tests on20th June 2017. Written informed consent was obtained from the parents or legal guardians of the adolescents and from the patients themselves, prior to study enrolment, following Helsinki declaration.

Whole blood was collected from the two patients by venipuncture in vacutainer tubes containing EDTA (Becton Dickinson, Franklin Lakes, NJ, United States) at the time of the diagnosis. After 6 months of treatment, blood was drawn again (M on 16th November 2017, G on 12th December 2017).

### 2.2. DNA Extraction and HIV-1 Amplification

Both patients, G and M, were analyzed at two different time points, before (16th and 20th June 2017, respectively) and after the start of antiretroviral therapy (16th November and 12th December 2017, respectively). Total DNA was extracted from the whole blood by DNA purification using Maxwell^®^ (Promega, Madison, WI, United States). The HIV-1 protease, RT, V1V2 and gp41 fragments were amplified from DNA using HIV genotype-specific primers (Appendix A). PCR was carried out using GoTaq^®^ (Promega). All amplicons were purified using the QIAquick PCR purification kit (Qiagen, Hilden, Germany).

### 2.3. Next-Generation Sequencing and Data Analysis

Library preparation for Illumina sequencing was done using a Nextera^®^ XT DNA sample preparation and index kit (Illumina, San Diego, CA, United States) according to the manufacturer’s manual. Resulting libraries were normalized and pooled for subsequent sequencing on an Illumina MiSeq platform using the 2 × 150 cycle paired-end sequencing protocol.

Results were mapped and aligned to the HXB2 reference. The consensus sequence for the reads obtained from the individual samples was generated by Geneious software [39].

The BBMerge tool was used to merge the two overlapping paired reads into a single longer read, and reads shorter than 60 bp were deleted. An in-house pipeline was used to filter identical reads in the same sample. This approach allows limitation of PCR bias and recombination. However, it could affect the number of reads per template resulting in reduced coverage [40].

### 2.4. Subtyping of Protease and RT Sequences

Protease–RT and integrase sequences generated by resistance testing were trimmed to equivalent lengths (1300 and 864 bp, respectively) and aligned with representative sequences available in the Los Alamos database (http://hiv-web.lanl.gov; accessed on 27 February 2021) using the CLUSTAL algorithm implemented in BioEdit version 7.0.4 (http://www.mbio.ncsu.edu/BioEdit/page2.html; accessed on 27 February 2021). Two to five strains representative of each of the nine pure subtypes (A–D, F–H, J, and K) and all known circulating recombinant forms (CRFs) were chosen as references.

Subtype assignment was performed using the MEGAX program [41] and the maximum-likelihood phylogenetic approach with bootstrapping on 1000 replicates.

The SimPlot software [42] and SplitsTree program [43] were used to check for possible recombination.

### 2.5. Phylogenetic Datasets

For each amplified region three different datasets were built, the first only encompassed reads obtained from samples collected before treatment (*n* = 193, *n* = 582, *n* = 288, *n* = 82 for protease, RT, V1V2 and gp41, respectively), the second including reads from samples during treatment (*n* = 145, *n* = 774, *n* = 226, *n* = 117 for protease, RT, V1V2 and gp41, respectively), and the third was composed of reads of samples collected before and during treatment (*n* = 298, *n* = 608, *n* = 513, *n* = 166 for protease, RT, V1V2 and gp41, respectively).

For dated phylogeny a distinct dataset was built including additional strains obtained from patients residing in the metropolitan area of Milan with the same subtype (*n* = 57).

To limit the bias due to APOBEC signatures, each of the APOBEC-context mutations were examined for their prevalence in ART-naïve and ART-experienced viruses according to their association with stop codons (W = >*). Reads were deleted and distance, phylogenetic and phylodynamic analyses were repeated. The final dataset included 185, 559, 288 and 76 reads for protease, RT, V1V2 and gp41, respectively, from untreated samples; 137, 720 and 226 for the same regions from samples during treatment and 287, 599 and 513 reads from samples obtained before and after treatment.

### 2.6. Phylogenetic and Phylodynamic Analyses

The Bayesian phylogenetic tree was reconstructed by means of both MrBayes and Beast using the general time-reversible (GTR)+ gamma distribution (G)+ proportion of the invariable sites (I) model, which was selected as the best-fitting evolutionary model using the JmodelTest [44]. A Markov chain Monte Carlo (MCMC) search was made for 1–17 × 106 generations according to different datasets using tree sampling every 100th generation and a burn-in fraction of 50%. Statistical support for specific clades was obtained by calculating the posterior probability (pp) of each monophyletic clade, and a posterior consensus tree was generated after a 50% burn-in. Clades with a pp > 0.7 were considered significant.

To evaluate if the sequence clustering was influenced by the application of a molecular clock, maximum-likelihood trees were also built using IQTree V.2 with 1000 bootstrap replicates and the previous selected evolutionary model [45].

The dated tree was built using Beast v.1.8.4 [46] software using the previous selected model and a previous estimated evolutionary rate [47]. Different coalescent priors were tested using both strict and relaxed molecular clock models [48]. BF calculations were performed with Tracer v.1.6 software (http://beast.bio.ed.ac.uk/Tracer; accessed on 27 February 2021). Convergence was assessed based on the effective sample size (ESS) value and only parameters with estimates with ESS > 300 were accepted. The maximum clade credibility (MCC) tree was then selected from the posterior tree distribution using TreeAnnotator v.1.8.4 available in the BEAST software package.

The MEGA X program was used to evaluate the genetic distance between and within sequences of each dataset [41]. A root-to-tip regression analysis was performed using TempEst in order to investigate the temporal signal of the dataset and the genetic distances from a consensus sequence of the naïve time-point sequences obtained during therapy [49].

## 3. Results

### 3.1. Clinical Profile

The blood and urine culture returned a negative result for HSV1-2, CMV, toxoplasmosis, HCV, parvovirus, and *Mycoplasma pneumonia*. The serology for EBV showed the presence of a previous infection. The QuantiFERON test result was negative. The HIV test was performed both on anti-HIV specific antibodies and HIV antigens. The test for HIV1/2-Ab/Ag was positive. There were 9,404,000 copies/mL of HIV-RNA and the CD4+ T lymphocyte count was 438/mm^3^ (12.8%). Therefore, the diagnosis of HIV was confirmed and disclosed to M and her parents. A drug resistance test was then performed, and its result was negative. A four-drug antiretroviral therapy based on dolutegravir, abacavir and lamivudine (in a single-tablet regimen) in addition to a single tablet of darunavir 800 mg/ritonavir 100 mg was started. After three weeks of antiretroviral therapy, HIV-RNA in the blood was at 514 copies/mL and the CD4+ T lymphocyte count was 820/mm^3^ (41%). Darunavir/ritonavir was interrupted after three months, given the decrease in viral load. The therapy was successful and after six months HIV-RNA had < 37 copies/mL with a CD4+ T lymphocyte count of 888/mm^3^ (39%).

HIV-RNA baseline viremia in G was 80,240 copies/mL with a CD4+ T lymphocyte count of 340/mm^3^ (16.8%). Upon drug resistance testing, which returned a negative result, the recommended therapeutic regimen was based on emtricitabine, tenofovir alafenamide and darunavir/c. The HCV antibody and HBs antigen tests were negative; HIV1/2-Ab/Ag were positive, as confirmed by Western blot analysis. After four weeks, HIV-RNA had 790 copies/mL and the CD4+ T lymphocyte count was 466/mm^3^ (28%). After six months, HIV-RNA had < 37 copies/mL and CD4+ T lymphocyte count was 838/mm^3^ (35%).

### 3.2. Subtypes and Genetic Distances

The HIV-1 protease, RT, V1V2, and gp41 regions were amplified and sequenced. The phylogenetic trees confirmed that the viral subtype present in G and his partner was a B subtype. There was no evidence of viral recombination observed in any portion. The lowest intrapersonal genetic distances were observed at baseline, before initiation of therapy, and in particular in the V1V2 fragment (M = 0.012 and G = 0.011) compared to samples during treatment at six months (distances ranging from 0.102 to 0.148). On the contrary, in the gp41 region a lower genetic distance was observed in samples during treatment at six months (M = 0.062 and G = 0.067) compared to the naïve samples (M = 0.113 and G = 0.129). Regarding the intra-patient genetic distances between the two time points, the lowest values were observed in the V1V2 fragment (M = 0.071 and G = 0.077) while a very high divergence was observed in the gp41 portion (M = 1.292 and G = 1.970). Appendix A reported genetic distances in different datasets considering or excluding APOBEC signatures. Root-to-tip regression analysis results are shown in Appendix A, also reporting the correlation coefficient and the coefficient of determination (R2). This analysis supports the overall low diversity of intra-patient sequences, reflecting the sampling during acute infection and the early start of treatment.

### 3.3. Variant Detection

The tree obtained by analyzing all of the reads achieved from the two patients before the beginning of the therapy showed the presence of a single major variant in the protease (Figure 1A) and RT regions (Figure 1B). The tree topology showed a toothcomb shape, with a low-grade variability. Three variants were detected for M and one for G in the V1V2 tree (Figure 1C). M’s variants stood as an outgroup of the entire tree, presenting one large clade (pp = 0.8) encompassing the most represented variant carried by the patients, and a small significant cluster (pp = 0.86) corresponding to a distinct M variant. In the gp41 tree (Figure 1D), G’s variant acted as an outgroup of the tree, which showed a main clade (pp = 0.83) with a common variant among the patients, from which a second large cluster originated (pp = 0.91) with a very different variant carried by both subjects. Analysis of the protease region during treatment showed the presence of a large common variant and a distinct cluster that only included an M variant (in blue in the Figure 2A). Similarly, the tree of the RT region (Figure 2B) presented a prevalent equal variant in the two patients. Two small clusters could be defined, the former (pp = 0.89) encompassing a separate M variant and the latter (pp = 0.7) a second common variant. In the V1V2 region, beside the common main variant, seven small clusters were identified (Figure 2C), four of which were minor distinct variants shared by both patients (pp = 1, pp = 0.83, pp = 0.79, pp = 0.72), two variants were found in G only (pp = 0.98, pp = 0.84) and one in M only (pp = 0.78).

Only one major common variant could be observed in the tree of the gp41 region (Figure 2D). When all reads obtained from naïve and cART treated samples were analyzed together, only a large common variant was observed in the protease, RT and V1V2 regions without any distinction between the time points (Figure 3A–C). On the contrary, in the gp41 tree (Figure 3D) we distinguished two main variants, one including only reads from patients during treatment, and a separate cluster (pp = 1) mainly encompassing the naïve variants. The analyses were repeated using the same datasets, and those cleaned for APOBEC signatures with the maximum likelihood approach obtained comparable results (Appendix A). No re-infection or viral recombination has been observed over time with the couple in the study.

### 3.4. Dated Tree Reconstruction

Comparison of the different coalescent models using the BF test on the entire dataset showed that the model best fitting the data was a coalescent prior BSP (2lnBF constant vs. BSP = 88.1, and exponential vs. BSP = 152.4) under a log-normal relaxed clock (2lnBF strict vs. relaxed clock = 54.8). In the tree, all sequences clustered significantly together (pp = 1) with a G sequence obtained that arose externally. The dated tree indicated a TMRCA for the cluster encompassing all patient isolates at 22.25 months (95%HPD: 8–43.6 months) corresponding to February 2017. The node related to the M sequence dated 19.3 months (95%HPD: 6–20 months) corresponding to April 2017 (data not shown).

## 4. Discussion

During the last decade, the knowledge that HIV establishes new infections mostly through one single variant has been supported by a growing amount of evidence [11,50,51]. It is now widely accepted by the scientific community that just one or a few variants are indeed able to be transmitted through the bottleneck of the mucosa, founding the new infection.

We performed phylogenetic analyses in a young couple who became infected through sexual activity using a next-generation sequencing (NGS) approach (Miseq platform); we analyzed the HIV-1 quasispecies present in samples and verified their evolution under treatment pressure. This event allowed us the unprecedented opportunity to study the T/F variants in a therapy-naïve couple and to verify the effect of antiretroviral therapy on viral evolution. The results showed the presence of a limited number of variants, ranging from one to two, which were transmitted from G to M.

We chose to analyze HIV proviral DNA, which is the “archived” compartment, since we believe it is more informative than plasma. By sequencing different HIV-1 regions (protease, RT, V1V2 and gp41), we observed that the average genetic distance between the viruses present in the two partners increased over time, as already reported in different studies, with the exception of the gp41 region where treatment halved the viral genetic distance. In samples obtained before and six months after the initiation of therapy, the pol region appeared to be highly conserved, as expected, compared to other analyzed portions, and only one major T/F variant seemed to be present. The env region was found to be highly variable, as expected, in particular in the gp41 portion, highlighting the presence of three common variants. The evolution of HIV-1 quasispecies was evaluated considering only viral variants that were present after six months of therapy. Upon antiretroviral treatment, one HIV-1 single variant was clearly dominant, although some minor variants could be observed in all considered portions, with the exception of the gp41 region. These minor variants were not visible at baseline probably due to the overwhelming presence of the main variants. Apparently, the segregation of variants sampled before and after therapy involved the gp41 region, but not the V1V2 gp120 region. None of the employed therapeutic approaches target this region, but, as it is an immunogenic region, it is subdued to selective pressures, nonetheless. A possible explanation for this apparent discrepancy is that the dominating V1V2 observed variant was previously completely obscured by a different variant.

In order to better understand the differences observed at the two evaluated time points, an additional analysis was conducted in which all the variants were included. When all the data from the two time points were pooled together, results showed that all the variants were already present at the time of transmission, with the exception of gp41. In fact, two variants were clearly identifiable among both the naïve and the post-therapy variants. Most likely, these two variants that were observed in all datasets can be considered the early T/F variants. This could be attributed to the sequestration of the virus in reservoir cells that were infected early in the course of infection, but were subsequently maintained at low levels in circulation as latently infected cells.

In agreement with the literature, we observed that the viruses in the newly infected subject were closely related to viruses present in the donor; this is in sharp contrast with results obtained when transmission occurs in a late stage of the infection [52,53].

It has recently become clear that the transmitted/founder virus is rarely the dominant variant in the donor, suggesting that the transmission bottleneck is not entirely stochastic but also involves the selection of specific viral phenotypes [54]. These findings were confirmed by data obtained from the SIV macaque model, and led to the hypothesis that the transmitted viral strains are sequestered in a long-lived reservoir during the early stage of infection, persist at a low level in the serum, and are preferentially selected for transmission [55].

A strong point of this work is the short timeframe between virus transmission and sampling; however, further longitudinal analysis of couples in which the recipient is sampled close to the acute stage of disease will be needed to better determine the selection of variants, and the relation among ancestral genotypes and variants, found in newly infected subjects.

The high homogeneity of quasispecies between the donor and the recipient in this study was due to the short interval between transmission and sampling, and to the high viral load in the transmitter, which negatively correlates with genetic distance between donor and recipient variants [56].

Literature data reported, for whole blood, an increase in the average genetic distance with respect to the first sample. Usually overall, the average genetic distance between the viruses present in two partners increases over time; however, this remains less evident in the blood compared to the plasma compartment.

Our phylogenetic analyses not only confirmed the probability of transmission, but also allowed us to date this event with good approximation. In addition to the existence of an epidemiological relationship between the two patients, the dated tree supported the time point and the probable direction of transmission. In detail, our analysis confirmed that G acquired the infection before in early 2017, while M acquired it a few months after (around April of the same year).

This is the first time that phylogenetic analyses have been performed on an HIV-infected couple in both individuals at the therapy-naïve stage and upon cART. However, our study presented some limitations, the most relevant is intrinsic to the used sequencing methodology based on amplicons. The derived PCR bias could affect the interpretation of the viral variants. Understanding which HIV-1 variants are most likely to be transmitted is important for vaccine design and predicting virus evolution.

## Figures and Tables

**Figure 1 viruses-13-00513-f001:**
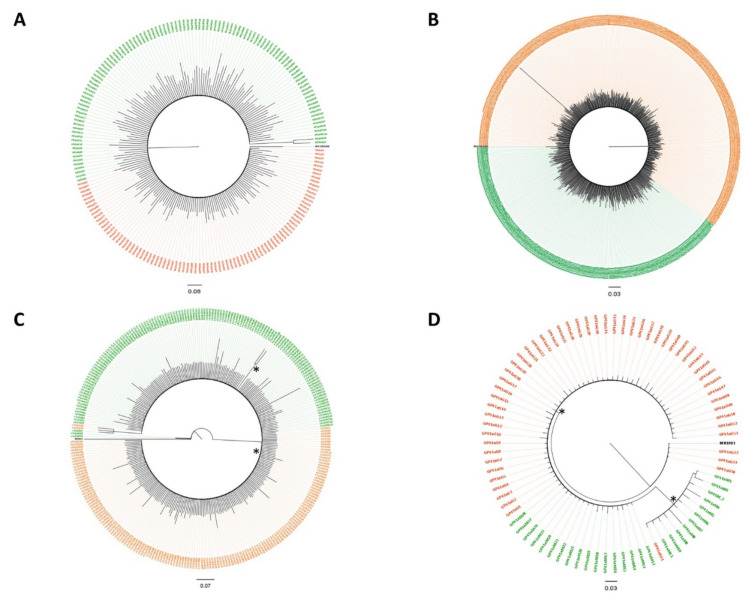
Bayesian trees for quasispecies of the human immunodeficiency virus (HIV) protease (**A**), RT (**B**), V1V2 (**C**) and gp41 (**D**) regions from the male (G, orange color) and female (M, green color) partners of the couple based on the analyses of whole-blood samples obtained before the start of therapy. Significantly different clusters are indicated with an asterisk.

**Figure 2 viruses-13-00513-f002:**
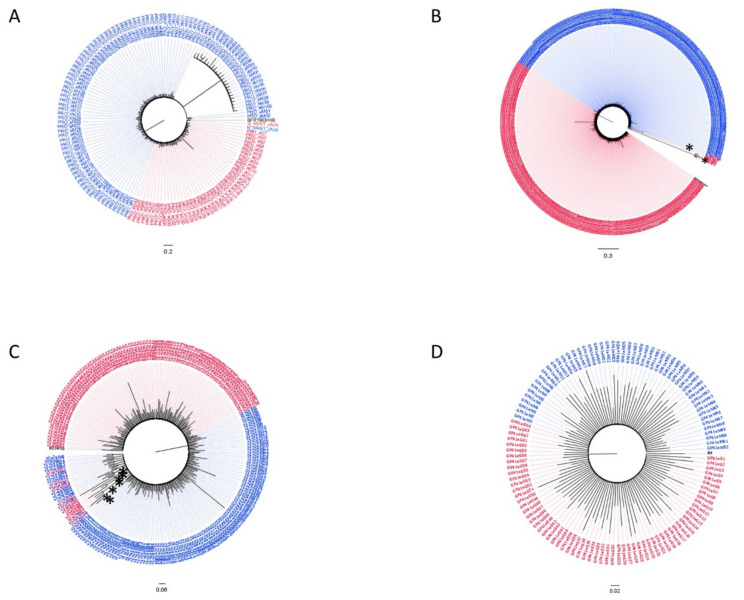
Bayesian trees for quasispecies of the HIV protease (**A**), RT (**B**), V1V2 (**C**) and gp41 (**D**) regions from the male (G, red color) and female (M, blue color) partners of the couple based on the analyses of whole-blood samples obtained during treatment. Significantly different clusters are indicated with an asterisk.

**Figure 3 viruses-13-00513-f003:**
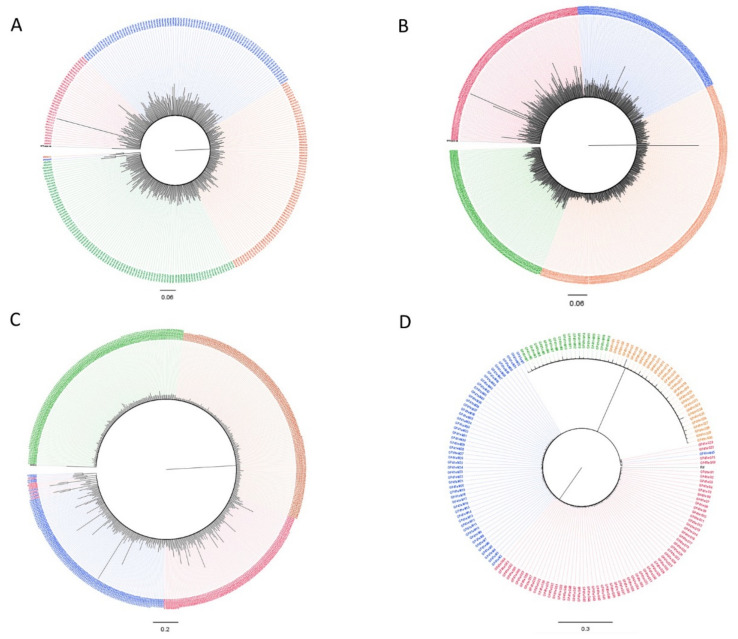
Bayesian trees for quasispecies of the HIV protease (**A**), RT (**B**), V1V2 (**C**) and gp41 (**D**) regions from the male (G) and female (M) partners of the couple based on the analyses of whole-blood samples obtained before the start of therapy (orange and green color for G and M, respectively) and during treatment (red and blue color for G and M, respectively).

## Data Availability

Data will be available upon reasonable request.

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
