# Peer review of "Early-Transmitted Variants and Their Evolution in a HIV-1 Positive Couple: NGS and Phylogenetic Analyses"

_viruses, 2021, doi:10.3390/v13030513_

Round 1

Reviewer 1 Report

Lai and colleagues describe a transmission pair of adolescents diagnosed during acute/recent HIV-1 infection. They conduct HIV-1 DNA population analyses from short-read sequencing of 4 HIV-1 sub-genomic regions. They had two time points available: one at diagnosis and one after 6 months of successful antiretroviral treatment. Thanks to their efforts, they were able to date the estimated time of HIV-1 acquisition and transmission, and investigate transmitter/founder viruses in the transmission pair and if they were still present after treatment introduction. The case report is definitely interesting, as it is very difficult to identify transmission pairs during acute/early stages of infection. I find major limitations in the study, the interpretation of data, and how the data are presented (it is difficult for the reader to interpret the trees and follow the manuscript). I recommend to resolve, or at least address these limitations.

Major Issues:

-Language. The manuscript is well written. However, I strongly advice to revise the way the two study participants are addressed. “Boy”, “Girl”, “His girlfriend”. Similarly, words like “source of infection”, “heterosexual activity”, heterosexual behavior”, “unfortunate event”, “G…infecting M” should be revised. I recommend the following document from the NIAID addressing the use of affirming and not stigmatizing words surrounding People Living with HIV. https://www.hanc.info/Documents/NIAID%20HIV%20Language%20Guide%20071520%20revised.pdf

-The analysis of longitudinal samples can be very informative. In this manuscript, however, is not always clear when the timepoints are located in time relative to the estimated time of infection and after the introduction of ART. I would recommend to clearly label the two time points with time from estimated day of detectable infection (EDDI, ref: https://www.ncbi.nlm.nih.gov/pmc/articles/PMC7078584/), Fiebig stage (if possible) or at least time from diagnosis, and time (months?) from ART introduction.

-Sequence analysis of HIV-1 variants was based on total DNA from whole blood. Thus, it includes both defective proviruses and those truly contributing to the replicating population during acute infection and the latent reservoir during ART. Despite the limitations of very short sub-genomic sequencing in parsing infectious proviruses, I would suggest to look for signatures of APOBEC3G/F editing and exclude those from distance, phylogenetic and phylodynamic analyses.

-Analysis of genetic distances: I suggest the Authors to complement their distance analysis with a root-to-tip analysis in which they calculate the distance of each sequence from the 2nd time point from a consensus of the sequences from the 1st time point.

-Line 234-235 and discussion. The authors should expand on the explanation for the apparent segregation of variants sampled before and after therapy exclusively in the gp41 data set. Why V1V2 did not provide a similar result? What are the chances that, in both individuals, the dominating variant during untreated infection was completely obscured by a different variant?

In addition, in Fig.1-D I see two variants, however, in the combined dataset in Fig.3-D I see only one variant from the 1st time point. I suggest to complement the Bayesian time-based tree with a Maximum Likelihood or Neighbor-Joining tree. This would rule out whether the clustering to sequences is due to the enforcement of a molecular clock. Please refer to https://pubmed.ncbi.nlm.nih.gov/28891813/ for additional details.

-The authors generated libraries of HIV-1 DNA sequences without the use of molecular barcodes that allow to control for PCR bias and recombination (ref: https://pubmed.ncbi.nlm.nih.gov/26041299/). To partially resolve this issue, they collapsed all identical sequences. I would recommend the authors to mention that their approach has the limitation of not preserving quantitation (in the methods and/or in the discussion).

-Line 69-79. I appreciate the comparison that the Authors make regarding clonal dynamics in HIV-1 and HTLV. However, I would have chosen more appropriate references on the subject of HTLV-1 persistence through clonal expansion. Seminal work from Bangham and colleagues should be included. https://www.sciencedirect.com/science/article/pii/S1044579X13001247?via%3Dihub

-Line 71. The Author cite one publication claiming an increase in reservoir size in 25% of individuals. The study from Bachmann and colleagues was based on a PCR-based assay that measures HIV-1 DNA, not the reservoir. New evidence, based on an assay that discriminate intact vs defective DNA fails to reproduce such a high fraction of cases in which the infectious proviruses increase over time (Peluso et al JCI-insight 2020, Gandhi et al JID 2020, Antar et al JCI 2020). I would revise and discuss these publications.

-Line 77-78. I advise the authors to include evidence demonstrating that non suppressible low-level viremia can be the result of virus production by clonally expanded replication competent proviruses; in these cases, viremia is not associated with replication, nor increased diversity.

Minor Issues:

-Line 64. Bllod should be blood

-Line 91. I find “Stunningly” superfluous.

-Line 98. “Drown”, should be “drawn”.

-Line 150. “Naïve samples” is confusing. Please revise.

-Line 177. “slight increase”, please clarify.

-Line 181-183. If western blot data are available (as for G), please include them; it would help with an even more accurate staging of HIV-1 infection (Fiebig). “HIV test was performed and it resulted positive” please clarify which test was performed.

-Line 185. “Lamivudina” should be “Lamivudine”.

-Line 185bis. “a tablet of Darunavir/r” please clarify regimen dosage.

-Line 193. “Tenofovir, Alfenamide” should read “tenofovir alfenamide”.

-Line 199. Please clarify that the authors amplified HIV-1 DNA from whole blood.

-Line 214. “Without high variability among reads”, please clarify.

Author Response

Response to Reviewer 1 Comments

Lai and colleagues describe a transmission pair of adolescents diagnosed during acute/recent HIV-1 infection. They conduct HIV-1 DNA population analyses from short-read sequencing of 4 HIV-1 sub-genomic regions. They had two time points available: one at diagnosis and one after 6 months of successful antiretroviral treatment. Thanks to their efforts, they were able to date the estimated time of HIV-1 acquisition and transmission, and investigate transmitter/founder viruses in the transmission pair and if they were still present after treatment introduction. The case report is definitely interesting, as it is very difficult to identify transmission pairs during acute/early stages of infection. I find major limitations in the study, the interpretation of data, and how the data are presented (it is difficult for the reader to interpret the trees and follow the manuscript). I recommend to resolve, or at least address these limitations.

 We would like to thank the Reviewer 1 for the overall positive assessments of the manuscript and for the forthcoming comments, which definitively contributed to an ameliorated and clearer version of many sections of the manuscript.

Major Issues:

-Language. The manuscript is well written. However, I strongly advice to revise the way the two study participants are addressed. “Boy”, “Girl”, “His girlfriend”. Similarly, words like “source of infection”, “heterosexual activity”, heterosexual behavior”, “unfortunate event”, “G…infecting M” should be revised. I recommend the following document from the NIAID addressing the use of affirming and not stigmatizing words surrounding People Living with HIV. https://www.hanc.info/Documents/NIAID%20HIV%20Language%20Guide%20071520%20revised.pdf

We are thankful to the Reviewer for such suggestion, which definitively points out an often-underestimated topic: words matter! We actually feel sorry for not considering appropriately the issue, which is now addressed as follows:

  • Line 15 and 269: “sexual behaviour” instead of “heterosexual”.
  • Line 95: “A fourteen-year-old female (M) was admitted” instead of “girl”.
  • Line 99: “M reported having a sexual relationship with a fifteen-year-old male (G)” instead of “boy”.
  • Line 99: the sentence “indicating him as the putative source of infection” was removed, as unnecessary and potentially offensive.
  • Line 199: “in G and his partner was” instead of “girlfriend”.
  • Line 270: “unfortunate” was removed.
  • Line 274: “which were transmitted from the G to the M” instead of “boy…girl”.
  • Line 321: “G… transmitting HIV to M”.

-The analysis of longitudinal samples can be very informative. In this manuscript, however, is not always clear when the timepoints are located in time relative to the estimated time of infection and after the introduction of ART. I would recommend to clearly label the two time points with time from estimated day of detectable infection (EDDI, ref: https://www.ncbi.nlm.nih.gov/pmc/articles/PMC7078584/), Fiebig stage (if possible) or at least time from diagnosis, and time (months?) from ART introduction.

We agree with the Reviewer. Information about the estimated date of infection, which are indeed confirmed by the sequence analyses, are now clearly stated throughout the method and result sections. An accurate Fiebig staging was not possible to be assessed and therefore not reported. However, additional information on the HIV1/2-Ag/Ab clinical test is now mentioned at line 200 and 213 of the result section.

-Sequence analysis of HIV-1 variants was based on total DNA from whole blood. Thus, it includes both defective proviruses and those truly contributing to the replicating population during acute infection and the latent reservoir during ART. Despite the limitations of very short sub-genomic sequencing in parsing infectious proviruses, I would suggest to look for signatures of APOBEC3G/F editing and exclude those from distance, phylogenetic and phylodynamic analyses.

In agreement with the reviewer suggestion, supplementary analyses were added in the methods and results section. Lines 158-162 in the methods section now read as follow: “To limit the bias due to APOBEC signatures, each of the APOBEC-context mutations were examined for their prevalence in ART-naïve and ART-experienced viruses according to their association with stop codons (W=>*). Reads were deleted and distance, phylogenetic and phylodynamic analyses were repeated. The final dataset included 185, 559, 288, 76 reads for protease, RT, V1V2 and gp41, respectively, from untreated samples; 137, 720, 226 and for the same regions from samples during treatment and 287, 599, 513, reads of samples obtained before and after treatment.” Lines 222-223 in the result section now read as follow: “Supplementary Tables 1 and 2 reported genetic distances in different dataset considering or excluding APOBEC signatures”. The relative tables were added as supplementary files.

-Analysis of genetic distances: I suggest the Authors to complement their distance analysis with a root-to-tip analysis in which they calculate the distance of each sequence from the 2nd time point from a consensus of the sequences from the 1st time point.

Root-to-tip analysis was performed, as suggested. These supplementary analyses were added in the methods and results section. Lines 183-186 in the methods section now read as follow: “The MEGA X program was used to evaluate the genetic distance between and within sequences of each dataset [37]. A root‐to‐tip regression analysis was performed using TempEst in order to investigate the temporal signal of the dataset and genetic distances from a con-sensus sequences of the naïve time point to sequences obtained during therapy [45].”. Lines 223-225 in the results section now read as follow: “Root‐to‐tip regression analysis results are shown in Supplementary Figures 1 and 2 also reporting the correlation coefficient and the coefficient of determination (R2)”. The relative figures were added as supplementary files.

-Line 234-235 and discussion. The authors should expand on the explanation for the apparent segregation of variants sampled before and after therapy exclusively in the gp41 data set. Why V1V2 did not provide a similar result? What are the chances that, in both individuals, the dominating variant during untreated infection was completely obscured by a different variant?

The discussion section was expanded, according to the Reviewer’s comment, in order to disclose any potential concern (lines 308-313).

In addition, in Fig.1-D I see two variants, however, in the combined dataset in Fig.3-D I see only one variant from the 1st time point. I suggest to complement the Bayesian time-based tree with a Maximum Likelihood or Neighbor-Joining tree. This would rule out whether the clustering to sequences is due to the enforcement of a molecular clock. Please refer to https://pubmed.ncbi.nlm.nih.gov/28891813/ for additional details.

Following the Reviewer’s suggestion, all the analyses were run with maximum likelihood approach considering original datasets and those obtained after APOBEC cleaning. These supplementary analyses were added in the methods and results section. Figures were added as supplementary files.

-The authors generated libraries of HIV-1 DNA sequences without the use of molecular barcodes that allow to control for PCR bias and recombination (ref: https://pubmed.ncbi.nlm.nih.gov/26041299/). To partially resolve this issue, they collapsed all identical sequences. I would recommend the authors to mention that their approach has the limitation of not preserving quantitation (in the methods and/or in the discussion).

We now mentioned this limitation in the methods section at lines 132-135, which reads: “An in-house pipeline was used to filter identical reads in the same sample. This approach allows to limit the PCR bias and recombination. However, it could affect the number of reads per template resulting in reduced coverage”.

-Line 69-79. I appreciate the comparison that the Authors make regarding clonal dynamics in HIV-1 and HTLV. However, I would have chosen more appropriate references on the subject of HTLV-1 persistence through clonal expansion. Seminal work from Bangham and colleagues should be included. https://www.sciencedirect.com/science/article/pii/S1044579X13001247?via%3Dihub

We are thankful to the Reviewer for his/her appropriate suggestion. The reference is now cited as n. 26.

-Line 71. The Author cite one publication claiming an increase in reservoir size in 25% of individuals. The study from Bachmann and colleagues was based on a PCR-based assay that measures HIV-1 DNA, not the reservoir. New evidence, based on an assay that discriminate intact vs defective DNA fails to reproduce such a high fraction of cases in which the infectious proviruses increase over time (Peluso et al JCI-insight 2020, Gandhi et al JID 2020, Antar et al JCI 2020). I would revise and discuss these publications.

The raised issue was revised in the introduction section as follows: (lines 73-76) “Recent data showed that in treated HIV-1 infected subjects intact and defective provirus declined over time with a more rapid decline in the first years of ART and a faster decay in cells with intact genomes [28-30]”.

-Line 77-78. I advise the authors to include evidence demonstrating that non suppressible low-level viremia can be the result of virus production by clonally expanded replication competent proviruses; in these cases, viremia is not associated with replication, nor increased diversity.

The following sentence was added to clear the above-mentioned issue, together with an appropriate reference. “The identification of a highly expanded clone of HIV-infected CD4+ T cells produced infectious virus at a level that caused persistent plasma viremia. This result shows that cells containing replication-competent HIV-1 proviruses can clonally expand and persist in vivo [23]”.

Minor Issues:

-Line 64. Bllod should be blood

The typo is now corrected and the spelling checked throughout the manuscript.

-Line 91. I find “Stunningly” superfluous.

The word “stunningly” is now removed.

-Line 98. “Drown”, should be “drawn”.

The typo is now corrected.

-Line 150. “Naïve samples” is confusing. Please revise.

We agree with the Reviewer that such wording might be confusing. In order to disambiguate, the sentence (lines 150 … 152) now reads as follows: “obtained from samples collected before treatment … composed by reads of samples collected before and during treatment”.

-Line 177. “slight increase”, please clarify.

These markers usually increase in course of infection/inflammation and are considered as predictive of disease progression. In these cases, the increase was only minor and considered not worth of further consideration.

-Line 181-183. If western blot data are available (as for G), please include them; it would help with an even more accurate staging of HIV-1 infection (Fiebig). “HIV test was performed and it resulted positive” please clarify which test was performed.

The description of the diagnosis is now more detailed, as follows: “HIV test was performed both on anti-HIV specific antibodies and HIV antigens. Test for HIV-Ab/Ag resulted positive”. As an accurate Fiebig staging was not possible to be assessed, it is not reported.

-Line 185. “Lamivudina” should be “Lamivudine”.

The typo is now corrected.

-Line 185bis. “a tablet of Darunavir/r” please clarify regimen dosage.

The administered dosage is now specified and the sentence reads as follows: “in addition to a single tablet of Darunavir 800mg/Ritonavir 100mg”.

-Line 193. “Tenofovir, Alfenamide” should read “tenofovir alfenamide”.

The sentence now reads as follows: “tenofovir alfenamide”.

-Line 199. Please clarify that the authors amplified HIV-1 DNA from whole blood.

This information was clearly stated in the method section as follows: “Total DNA was extracted from whole blood by..”.

-Line 214. “Without high variability among reads”, please clarify.

The sentence was changed accordingly.

Reviewer 2 Report

This manuscript characterizes HIV-1 evolution in a heterosexual transmission pair, sampled at the time of presumed acute infection in the recipient female partner and at 6 months after the start of ART. The authors performed NGS and phylogenetic analysis of 4 HIV-1 regions derived from the proviral DNA. They postulate that the male donor was acutely infected at the time of transmission.

Major comments:

  1. The authors postulate that the dated tree reconstruction indicated the time and directionality of transmission; however, the TMRCA confidence intervals for the total and M sequences were very wide and overlapped considerably, therefore how sure can the authors be that G was the donor and M the recipient, and not the other way around? The clinical data do support the acute retroviral syndrome in M at the time of sampling, but were any serologic or other tests done on M that could help establish the timing of infection? This doubt is also because G had only one variant in the V1V2 region, while M had three.
  2. Did the authors compare the MRCA distances at baseline and on ART? This is important in order to assess the virus evolution (and thus replication) under therapy, which is an important (and controversial) subject in the HIV-1 research field. The viral diversity does not show a clear pattern: for one region, it is higher at ART than at baseline, whereas for another region it is the other way around.
  3. The data on the virus diversity, MRCA distances and other measures of virus evolution have to be shown as a table for all regions sequenced.

Minor comments:

  1. Lines 77-79: "Low-level viremia and replication burst during the blips result in HIV-1 genetic diversity and shape the pool of viral variants and the reservoir" - this is a very controversial statement, as it is still unclear whether blips reflect bursts of virus replication or simply virus production from stable reservoirs without infection of new cells.
  2. Lines 84-86: "approaches that limit the number and the heterogeneity of T/F variants could become promising therapeutic avenues" - it is unclear which approaches are meant.
  3. Lines 113-118 - this information is present twice (lines 102-107).
  4. Lines 206-207: "Regarding the interpersonal genetic distances between two time points" - unclear what is meant here.
  5. Lines 300-301: "In agreement with the literature, we observed that the viruses circulating in the newly infected subject were closely related to viruses present in the donor" - as proviral DNA and not plasma RNA was sampled, it is not possible to know which viruses were circulating and which were not.
  6. Lines 316-318: "the high viral load in the transmitter, which negatively correlates with genetic distance between donor and recipient variants" - no such correlation is shown. Or do the authors refer to the literature? Then a citation has to be provided.

Author Response

Response to Reviewer 2 Comments

This manuscript characterizes HIV-1 evolution in a heterosexual transmission pair, sampled at the time of presumed acute infection in the recipient female partner and at 6 months after the start of ART. The authors performed NGS and phylogenetic analysis of 4 HIV-1 regions derived from the proviral DNA. They postulate that the male donor was acutely infected at the time of transmission.

We are thankful to the Reviewer for his/her comments, which definitively lead to a substantial improvement of the manuscript.

Major comments:

  1. The authors postulate that the dated tree reconstruction indicated the time and directionality of transmission; however, the TMRCA confidence intervals for the total and M sequences were very wide and overlapped considerably, therefore how sure can the authors be that G was the donor and M the recipient, and not the other way around? The clinical data do support the acute retroviral syndrome in M at the time of sampling, but were any serologic or other tests done on M that could help establish the timing of infection? This doubt is also because G had only one variant in the V1V2 region, while M had three.

We postulated that the dated tree confirmed the direction of infection because G resulted external to clade showing a TMRCA that dated back those of M. Also, the epidemiologic inquiry is confirmed by data tree results. In addition, the clinical data are now better detailed (lines 198-217) and read as follow:

 “Blood and urine culture resulted negative for HSV1-2, CMV, Toxoplasmosis, HCV, Parvovirus, Mycoplasma pneumonia. Serology for EBV showed the presence of a previous infection. QuantiFERON test resulted negative. HIV test was performed both on anti-HIV specific antibodies and HIV antigens. Test for HIV-Ab/Ag resulted positive. HIV-RNA was 9,404,000 copies/ml and CD4+ T lymphocytes count was 438/mm3 (12.8%). Therefore, the diagnosis of HIV infection was confirmed and disclosed to M and her parents. Drug resistance test was then performed, and it resulted negative. A 4-drug antiretroviral therapy based on Dolutegravir, Abacavir and Lamivudine (in single tablet regimen) in addition to a single tablet of Darunavir 800mg/Ritonavir 100mg was started. After three weeks of antiretroviral therapy, HIV-RNA in blood was 514 copies/ml and CD4+ T lymphocytes count was 820/mm3 (41%). Darunavir/r was interrupted after three months, given the decreasing of viral load. Therapy was successful and after six months HIV-RNA was < 37 copies/ml with a CD4+ T lymphocytes count was 888/mm3 (39%). HIV-RNA baseline viremia in G was 80,240 copies/ml with a CD4+ T lymphocytes count of 340/mm3 (16.8%). Upon drug resistance testing, which resulted negative, the recommended therapeutic regimen was based on Emtricitabine, tenofovir alafenamide and Darunavir/c. HCV antibodies and HBs antigen resulted negative; HIV antibodies were positive, as confirmed by Western blot test. After four weeks, HIV-RNA was 790 copies/ml and the CD4+ T lymphocytes count was 466/mm3 (28%). After six months, HIV-RNA was < 37 copies/ml and CD4+ T lymphocytes count was 838/mm3 (35%).”.

  1. Did the authors compare the MRCA distances at baseline and on ART? This is important in order to assess the virus evolution (and thus replication) under therapy, which is an important (and controversial) subject in the HIV-1 research field. The viral diversity does not show a clear pattern: for one region, it is higher at ART than at baseline, whereas for another region it is the other way around.

All the genetic distances are now better elucidated by adding a Supplementary tables reporting all datasets. In addition, a root to tip analyses was performed.

  1. The data on the virus diversity, MRCA distances and other measures of virus evolution have to be shown as a table for all regions sequenced.

All the genetic distances are now better elucidated by adding a Supplementary tables reporting all datasets.

Minor comments:

  1. Lines 77-79: "Low-level viremia and replication burst during the blips result in HIV-1 genetic diversity and shape the pool of viral variants and the reservoir" - this is a very controversial statement, as it is still unclear whether blips reflect bursts of virus replication or simply virus production from stable reservoirs without infection of new cells.

We agree with the Reviewer about the complexity of the issue. The sentence is now tuned down and it reads as follows: “Low-level viremia and/or replication burst during the blips might contribute to HIV-1 genetic diversity and shape the pool of viral variants and the reservoir.”

  1. Lines 84-86: "approaches that limit the number and the heterogeneity of T/F variants could become promising therapeutic avenues" - it is unclear which approaches are meant.

During a new infection, we have strong evidences that the T/F variants are a couple only. Moreover, the initial phase of HIV-1 infection and the initial heterogeneity are correlated to the progression to disease. The sentence was meant to speculate about the role of T/F variants, about the possibility of shared features among them, which could be targeted by future therapeutic or prevention strategies. This speculation was purposely left open, as currently there is no such a strategy aiming to this potential vulnerability, but it could be target of future strategies.

  1. Lines 113-118 - this information is present twice (lines 102-107).

We are sorry about that. The sentence “Ethics Committee approval was considered unnecessary because, under Italian law, it is only required in the case of prospective clinical trials of medical products for clinical use (Arts. 6 and 9 of Legislative Decree No. 211/2003).” Is now removed from lines 102-107.

  1. Lines 206-207: "Regarding the interpersonal genetic distances between two time points" - unclear what is meant here.

We agree with the Reviewer. The sentence is now revised as follows: “Regarding the intra-patient genetic distances between two time points…” (lines 226-227)

  1. Lines 300-301: "In agreement with the literature, we observed that the viruses circulating in the newly infected subject were closely related to viruses present in the donor" - as proviral DNA and not plasma RNA was sampled, it is not possible to know which viruses were circulating and which were not.

In agreement with the Reviewer, the above-mentioned sentence resulted slightly ambiguous. Therefore, it was re-worded as follows: “…the viruses in the newly infected…”.

  1. Lines 316-318: "the high viral load in the transmitter, which negatively correlates with genetic distance between donor and recipient variants" - no such correlation is shown. Or do the authors refer to the literature? Then a citation has to be provided.

We are thankful to the Reviewer for pointing this out. An appropriate citation is now added in as n. 56: Kariuki et al. Retrovirology (2017) 14:22 DOI 10.1186/s12977-017-0343-8.

Reviewer 3 Report

Authors report on phylogenetic analysis of virus strains from a very young couple.

Authors tend to overrate their study which is hampered at this stage with unknowns and confounders.

  1. do not use the word stunning in the intro
  2. explain  better how infection dates were calculated
  3. So this is not an acute infection? was serology complete?
  4. Why did both patients receive different treatment if virus strains were so similar; is this based on resistance data? Where are these data?
  5. The use of different treatment regimens is a confounded for following up evolution, authors should discuss the possible impact
  6. I wonder that similar phylogenetics studies have not been done before, I remember studies were entire infection networks were analysed

Author Response

Response to Reviewer 3 Comments

Authors report on phylogenetic analysis of virus strains from a very young couple.

Authors tend to overrate their study which is hampered at this stage with unknowns and confounders.

We are thankful to the Reviewer 3 for her/his overall forthcoming comments, which underline some issues that needed to be addressed. Also, we appreciated the practical frankness. Please, find below our rebuttal.

  1. do not use the word stunning in the intro

the word “stunningly” was removed and the sentence tuned down.

  1. explain  better how infection dates were calculated

The dates identified in the anamnestic investigation are now clearly stated in the patients’ description (section 2.1; lines 101 and 112) and in the section DNA extraction and HIV-1 amplification (section 2.2; lines 122-124). Such information was mirrored by phylogenetic analyses, as now described in the method section 2.6, lines 192-195 and in the result section 3.4, lines 267-270.

  1. So this is not an acute infection? was serology complete?

Regarding the female subject (M), we can estimate it was an acute infection, since her HIV viral load was greater than 9 million copies/ml and HIV1-2 Ag/Ab greater than 132.9 UI (nv <1). Regarding the male subject (G), the timing of infection is uncertain, since his HIV viral load was slightly more than 80.000 copies/ml. HIV 1-2 Ag/Ab resulted positive.

  1. Why did both patients receive different treatment if virus strains were so similar; is this based on resistance data? Where are these data?

Resistance tests showed no drug-resistance for either of the two subjects. However, the clinical conditions were different and required different approaches. Briefly, HIV viral load displayed by M was 9.404.000 copies/ml and she was symptomatic (fever, lymphadenopathy), therefore we chose ARV treatment containing 1 PI, 2 NUC and 1 INSTI (dolutegravir) in order to reduce HIV viral load as quickly as possible. Contrarily, HIV viral load displayed by G was 80.000 copies/ml and he was asymptomatic. We chose a PI containing regimen because of his poor compliance, not to favor drug-resistant viral strains development.

The negative results of the tests for drug resistance are now disclosed at lines 181 and 189, as suggested by the Reviewer.

  1. The use of different treatment regimens is a confounded for following up evolution, authors should discuss the possible impact

We are thankful to the Reviewer for raising this issue. As the first time-point was collected before therapy, this is not a concern. The second timepoint was collected after 6 months of therapy, which is considered having a limited impact. Moreover, env and gp41 regions are not subdued to the pharmacological pressure induced by the employed therapeutic regimens. The raised issue would have been more concerning if considering the reservoir.

  1. I wonder that similar phylogenetics studies have not been done before, I remember studies were entire infection networks were analysed

We understand the Reviewer’s concern, but our research relies on unprecedented novelties, as it is very difficult to identify transmission pairs during acute/early stages of infection. Indeed, just a few publications employed an NGS approach with such detailed level on four genomic regions of HIV coupled with a Bayesian inference analysis methods, but none of them was performed on a couple before and after therapy. So far, the transmitted founder (T/F) variants were so closely followed and so precisely characterized in the monkey model only. This is the first study performing a phylogenetic analysis of the T/F variants in a couple within few weeks from infection before and after the initiation of antiretroviral therapy with such deep resolution.

Round 2

Reviewer 1 Report

The Authors did a good job addressing the comments; the current version of the manuscript is improved in language, clarity, and technical aspects. I appreciate that the Authors included new analyses that corroborate their findings. I only have a couple of comments that need to be addressed (see below).

-Line 144: The authors added the reference from Zhou et al, as suggested. But rather than addressing the lack of primer IDs and the impact on representing viral populations quantitatively, they address reduce coverage. Please discuss the reference accordingly. I understand that filtering identical sequences reduces PCR bias (and coverage); but still, the problem of confident (quantitative) representation of viral populations remains. Any interpretation of what is "dominant" or distinct between the two time points and the two individuals is a reflection of what you amplify by PCR and survives your QC and filtering. 

-Line 246: I suggest to add here (or in the discussion) that the analysis of distances (and root-to-tip) support the overall low diversity of sequences, a reflection of sampling during acute infection and the early start of treatment, which both prevent the accumulation of diversity and divergence from T/F.

-Figures 1D and 3D (and respective supplementary materials). I appreciate that the authors repeated the analysis with a different approach (ML). I think it is useful, for the reader, to see sequences from a transmission pair being well-mixed. I think that the observations on the gp41 data are an interesting anomaly, and I encourage the authors in expanding further in the results and discussion sections. In figure 1D, I see 78 sequences, 12 from the smaller cluster and 66 from the remaining larger group. However, in Figure 3D, I can count only 50 sequences from pre-ART, grouping with GP41eM1. Are these two different, only partially overlapping, sets of sequences? Are these spanning different regions of GP41? It would be useful to address that, an additional explanation of such a striking switch in GP41 sequences could be technical, due to the different data sets.

The Authors argument that “sequestration of the virus in the reservoir” may explain the almost complete (GP41eM1) absence of the pre-ART sequences in the second time point. Even if this was true, what are the chances that the same exact effect occurs in both individuals? For both G and M you see viral quasi-species that then disappear upon treatment. I recommend the authors to discuss whether sampling and datasets could have generated this observation.

If instead this is a true biological finding, can the authors identify patterns of mutations between the early variant (now disappeared) and the dominant population persisting on therapy? To pinpoint the mutations that could explain the apparent selection of this variant between the two time points, a highlighter plot (LANL, HIV sequence database) could be useful. In the discussion, the authors reference immune pressure; have the authors looked for HLA restricted epitopes that could be under CTL selection?

Reviewer 2 Report

The authors addressed some comments, however I do not understand the new Supplementary Figures 1 and 2. Apparently, the goal is to show that the root-to-tip distances increase in time, but these correlation graphs make no sense. Why is there always only one dot at the first time point and a huge amount of dots at the second time point? Is the dot at the first time point an average value? In any case, correlation analysis is not suitable here, it is needed to use another method (like t-test or Mann-Whitney test) to compare the distances between the first and the second time points.

I also still do not understand the reasoning behind the directionality of infection. Given that the infection dates were likely very close, can only the phylogenetic analysis be used to postulate who infected who? Also, where is the dated tree shown? I cannot see in in the figures and it says "data not shown" in the text. If it is the only evidence the authors have for the direction of infection, the data has to be shown.

Reviewer 3 Report

My comments were adequately addressed

Round 3

Reviewer 1 Report

I have no further comments.  In case the Authors plan to do similar studies on intra-host viral populations in the future, I recommend to modify the experimental approach to obtain HIV sequences. These are very fascinating and important studies, thus it is key to get as much information as possible from such precious samples and their viral sequences. 

Reviewer 2 Report

I still have a problem with the directionality of infection. The authors added Supplementary figure 6 showing the dated tree, but this figure is very unclear. What is shown by all the unnamed branches? Also, as stated in the text and also now shown in the figure, the confidence intervals for the TMRCA distances overlap considerably (22.25 months (95%HPD: 8-43.6 months) and 19.3 months (95%HPD: 6-20 months)), how sure can the authors be of the direction of infection? The legend to the figure says "Dated tree showing statistically-significant support for clades along the branches" but what is the level of statistical significance? What test was used to calculate it? Given the wide overlap of 95% CIs, it is difficult to believe the difference could be significant.

Also, the authors say in their response that the evidence of the direction of infection is that "The strain of G is located outside the cluster as an outgroup and dated before the M strain" - but if one is looking carefully at the figure, the G-derived sequence that is an outgroup is designated "G_e" (which probably means "experienced") but the sequence "G_n" (which likely means "naive") is located inside the cluster. Therefore, the outgroup is formed by the later sequence from the therapy period, whereas the sequence from before therapy does not map outside the cluster. Hence, while I do not doubt there is a strong epidemiological link between the two patients, I do not at present see a proof of the direction of infection. The authors should either provide the evidence or remove the statements about the direction of infection from the manuscript.

By the way, the figure is not referred to in the text, the text still says "data not shown".